# An Andrographolide from *Helichrysum caespitium* (DC.) Sond. Ex Harv., (*Asteraceae*) and Its Antimicrobial, Antiquorum Sensing, and Antibiofilm Potentials

**DOI:** 10.3390/biology10121224

**Published:** 2021-11-24

**Authors:** Kokoette Bassey, Patience Mamabolo, Sekelwa Cosa

**Affiliations:** 1Pharmaceutical Sciences Division, School of Pharmacy, Sefako Makgatho Health Sciences University, Molotlegi Street, Ga-Rankuwa, Pretoria 0204, South Africa; Patience.mamabolo@smu.ac.za; 2Department of Biochemistry, Genetics and Microbiology, University of Pretoria, Private BagX20, Hatfield 0028, South Africa; sekelwa.cosa@up.ac.za

**Keywords:** andrographolide, antimicrobial, antiquorum sensing, antibiofilm, *Helichrysum caespititum*, multidrug resistance

## Abstract

**Simple Summary:**

Gonorrhea is a major public health concern globally and more than 800,000 new infections occur in the USA alone each year, according to the Centers for Disease Control (CDC). Health complications triggered by gonorrheal infection include HIV/AIDS and infertility, just to mention but a few. The Gram-negative gonococci bacteria resist known antibiotic treatments over the years, including penicillin, tetracycline, and fluoroquinolones, with only cephalosporins available for treatment currently. Resistance to the cephalosporins in many countries underlines the dire need for new antigonorrheal drugs. This study explored the potential of isolating an antigonorrheal compound from a nonpolar extract of *Helichrysum caespititium*. The 10-methyl-8-(propan-17-ylidene)naphthalen-9-yl)-11-vinyl-14-hydroxyfuran-16-one (**CF6**) isolated from the chloroform extract of the plant inhibited *Neisseria gonorrhoeae* with a MIC value of 60 µg/mL compared to ciprofloxacin with a MIC of 1 µg/mL. This compound indicates potential as a new antigonorrheal lead molecule.

**Abstract:**

*Helichrysum caespititium* (DC.) Sond. Ex Harv., (*Asteraceae*) is a medicinal plant indigenous to South Africa. Its non-polar extracts exhibit significant antimicrobial and, in particular, antigonorrheal activity. This study aimed at isolating and purifying the active antigonorrheal compound from its chloroform extract and validating its inhibition potential on quorum sensing (QS) and biofilm formation of multi-drug resistant (MDR) pathogens. Phytochemical investigation of aerial parts of *H. caespititium* afforded a diterpene lactone (**CF6**). The effect of **CF6** on violacein production and biofilm formation was studied using in vitro quantitative violacein inhibition (*Chromobacterium violaceum*) and biofilm formation (*Streptococcus pyogenes*, *Staphylococcus aureus*, *Escherichia coli*, *Klebsiella pneumoniae*, *Neisseria gonorrhoeae*, and *Pseudomonas aeruginosa*). The structure of **CF6** was characterized using FTIR, NMR, and UPLC-MS data accordingly, as 10-methyl-8-(propan-17-ylidene)naphthalen-9-yl)-11-vinyl-14-hydroxyfuran-16-one. The susceptibility testing of the pathogens against **CF6** revealed *Neisseria gonorrhoeae* was noticeably susceptible with a MIC value of 60 µg/mL, while *Streptococcus pyogenes* and *Staphylococcus aureus* showed MIC of 125 µg/mL. All gram-negative pathogens, *Escherichia coli, Klebsiella pneumonia* and *Pseudomonas aeruginosa* were inhibited at 250 µg/mL. **CF6** also inhibited the production of violacein by 51.88% at 250 µg/mL and prevented cell attachment by 40.76–81.18%, with *N*. *gonorrhoeae* being highly prohibited from forming biofilm. In conclusion, 10-methyl-8-(propan-17-ylidene)naphthalen-9-yl)-11-vinyl-14-hydroxyfuran-16-one is the first of its kind to be isolated from the non-polar (chloroform) extract of South African *Helichrysum caespititium* with antigonorrheal, antimicrobial, antiquorum sensing, and antibiofilm properties. The compound may serve as a drug candidate against MDR pathogens.

## 1. Introduction

*Helichrysum caespititium* (DC.) Sond. Ex Harv., (*Asteraceae*) has numerous biological activities [1,2,3,4,5,6,7,8]. Among these are antimicrobial and antioxidant activities. Coupled with the many traditional uses, the phyto-constituents in the plant are purportedly responsible for the biological activities and such uses. A comprehensive review by [9] highlights the different classes and the phytoconstituents’ types that have been isolated from different *Helichrysum* species from South Africa. The family of isolated compounds include phenolic derivatives, phloroglucinols, pyrones, diterpenes, triterpenes, flavonoids, chalcones, pyranchalcones, and flavanones, among others. A separate phytochemical investigation of aerial parts of *Helichrysum niveum* by [10] reported three new compounds ((1-benzoyl-3 (3-methylbut-2-enylacetate)-phloroglucinol or helinivene A, 1-benzoyl-3 (2*S*-hydroxyl-3-methylbut-3-enyl)-phloroglucinol or helinivene B, and 8-(2-methylpropanone)-3*S*,5,7-trihydroxyl-2,2-dimethoxychromane or helinivene C), and six known acylphloroglucinols (namely, 1-(2-methylbutanone)-4-*O*-prenyl-phloroglucinol, 1-(2-methylpropanone)-4-*O*-prennyl-phloroglucinol, 1-(butanone)-3-prenyl-phloroglucinol, 1-(2-methylbutanone)-3-prenyl-phloroglucinol, 1-butanone-3-(3-methylbut-2-enylacetate)-phloroglucinol, and 1-(2-methylpropanone)-3-prenylphloroglucinol), caespitate, and a known dialcohol triterpene called 3β-24-dihydroxyterexer-14-ene that were isolated from South Africa’s *H. niveum*.

However, only two compounds, caespitin [11] and 2-methyl-4-[2′, 4′, 6′-trihydroxy-3′-(2-methylpropanoyl)-phenyl] but-2-enyl acetate [12], have been isolated from the polar extracts of *H. caespititium.* From our study, a thin layer chromatography analysis of polar ethanol and chloroform extracts revealed that there are still other phytochemicals that should be isolated from this plant species. In addition to this observation, the non-polar extract of *H. caespititium* from our previous study [13] exhibited the best of the antigonorrheal and antioxidant activities. Hence, these formed the rational for this study to investigate the non-polar extract of *H. caespititium* with a view to isolating the antigonorrheal compound from the plant non-polar extract. In addition, we aimed to validate the compound’s biological activities, particularly antiquorum sensing (AQS) and antibiofilm potential against pathogens of concern such as *Neisseria gonorrhoeae*, *Escherichia coli*, *Klebsiella pneumoniae*, *Pseudomonas aeruginosa*, *Streptococcus pyogenes*, *and Staphylococcus aureus*.

According to the WHO [14], the above-mentioned pathogens are public health threats associated with the challenge of antibiotic resistance. A majority of these pathogens are biofilm forming. Biofilm formation is regarded as a virulence factor due to its contribution to the difficulty of treatment or eradication with antimicrobial treatment. Moreover, most bacteria control such virulence factors through a QS signal pathway, also known as cell-to-cell communication, whereby pathogens interact to form coordinated functions like formation of biofilm, sharing of nutrients, causing infection, and others [15]. Since plant-derived secondary metabolites are key to phytomedicine and are reported to possess versatility in the management of infectious diseases, we, therefore, conjectured that an andrographolide derived from *Helichrysum caespitium* comprises ability to disrupt bacterial growth, prevent the formation of biofilms, and act as an antipathogenic agent that inhibits the production of violacein by *Chromobacterium violaceum*.

## 2. Materials and Methods

### 2.1. Plant Material Collection and Identification

The *Helichrysum caespititium* (DC) Harv plant material was collected in Ga-Mamabolo (23.8346° S, 29.8844° E) Masealama, Mankweng district, Limpopo province, South Africa. The plant was identified by Indigenous Knowledge Systems (IKS) Practitioner and the South African National Biodiversity Institute (SANBI), Pretoria, and a voucher specimen with number HC01 was deposited in the School of Pharmacy of Sefako Makgatho Health Sciences University (SMU). The plant material (1.17 kg) was air-dried in the laboratory and, once dried, was ground to a fine powder using a mill (Kinematica AG, Lucerne, Switzerland) and stored until use.

### 2.2. Helichrysum Caespititium Extraction

About 1.05 kg of the powdered plant material was transferred to a 1000-mL beaker and 500 mL of chloroform was added. The mixture was sonicated for 30 min at a temperature of 25 °C. The mixture was then filtered using Whatman No. 1 filter paper. The process was repeated twice with the same plant material. The combined filtrate was pooled together and evaporated using a Stuart evaporator (Cole Parmer Ltd., Stone-England, UK) connected to a Vacuubrand MZ 2C NT pump (Vacuubrand GmBH + Co Kg, Wertheim, Germany) to afford 37.27 g of the chloroform extract.

### 2.3. Isolation from the Chloroform Extract

The non-polar dichloromethane (DCM) extract of H. caespititum was previuosly reported to exhibit antigonorrheal activity [16]. Our preliminary study indicated that the chlorofom was a better extractant in terms of the yield of the extract. As a result, we investigated the chloroform extract for the isolation of the active compound responsible for the antigonrrheal activity of the H. caespititum. About 37.27 g of H. caespititium chloroform extract concentrated in 50.0 mL of chloroform was adsorbed to 35.01 g of dry silica and allowed to dry. The plant extract–silica mixture was allowed to air dry prior to loading into a column for separation. A 30 mm od × 2.0 mm wall × 600 mm long for vacuum liquid chromatography (VLC) with B24 sockets and ground glass stoppers B24 with sintered disc P3 & PTFE S/C glass column (C.C. Imelmann (PTY) Ltd., Robertsham-Gauteng, South Africa) was mounted on a clamp support. The column was wet packed with a silica gel slurry prepared by adding 50 g of dry silica in 50 mL of acetone. The silica slurry was filled up to 65% of the length of the column. The residual solvent from the silica slurry was allowed to drip off the column until it was slightly above the packed silica gel. At this point, the dry silica–plant extract was loaded on the top of the wet silica gel. This was followed by the placement of a cotton wool above the loaded plant extract. About 10% of the column length was allowed for the addition of the mobile phase.

### 2.4. Column Chromatography

The column was eluted under the vacuum (VLC) as follows. The first batch elution of the column was done using about 1 L of the *n*-hexane sub-fraction. This was followed by the second and third batches of elution with dichloromethane (DCM) and chloroform (CHCl_3_) to afford the DCM and CHCl_3_ sub-fractions. The column was extruded with CHCl_3_:EtOAc (9:1 *v*/*v*; 1:1 *v*/*v*) and EtOAc. These three main sub-fractions (hexane, dichloromethane, and chloroform) were evaporated under pressure and, upon analysis by TLC CHCl_3_:EtOAc (9:1 *v*/*v*), the CHCl_3_ sub-fraction proved to contain more compounds and was further re-chromatographed. A separate column was packed using the same protocol described for the isolation of the compounds from the CHCl_3_ sub-fraction using CHCl_3_:EtOAc (9:1 *v*/*v*) as the mobile phase. Seven different pooled sub-fractions, labelled **CF1**–**CF7**, were collected and again analyzed by 1- and 2-dimension TLC to check for the purity of the isolated compounds.

### 2.5. UPLC-MS and NMR Instrumentation Used for Structural Elucidation

The isolated compounds were introduced by full-loop injection (1.0 μL) into a UPLC (Waters Acquity chromatographic system; Waters, Milford, MA, USA) equipped with a photodiode array detector, which was used to optimize the separations during the initial analyses. Pure compounds were separated on an Acquity UPLC BEH C_18_ column (150 mm × 2.1 mm, i.d., 1.7-μm particle size; Waters) maintained at 40 °C. The mobile phase consisted of 0.1% aqueous formic acid (Solvent A) and HPLC grade (Merck^TM^, Darmstadt, Germany) acetonitrile (Solvent B), at a flow rate of 0.3 mL/min. Gradient elution was applied as follow: 15% B to 35% B in 7 min, changed to 50% B in 1 min (held for 2.5 min), before returning to the initial ratio in 0.5 min (a total run time of 11 min). Data were managed by Markerlynx 4.1 chromatographic software.

The UPLC system was interfaced with a Xevo G_2_QT mass spectrometer (Waters, 34 Maple street, Milford, MA, USA). For the UPLC-QTOF-MS analyses, the same column, elution gradient, and flow rate were used as before. Although both positive and negative ionization modes were applied, the results obtained indicated that higher sensitivities and more information were obtained in the negative mode. The mass spectrometer was, therefore, operated in negative ion electrospray mode using nitrogen as the desolvation gas at a flow rate of 600 L/h. A desolvation temperature of 350 °C and a source temperature of 100 °C were used. The capillary and cone voltages were set to 2500 and 40 V, respectively. Data were collected in the range of *m*/*z* 100 to 1200.

### 2.6. Test Pathogens

The following pathogens were used in the present study: *Streptococcus pyogenes* ATCC 19615, *Staphylococcus aureus* ATCC 25923, *Escherichia coli* ATCC 10536, *Klebsiella pneumoniae* ATCC 33495, *Neisseria gonorrhoeae* ATCC 49981, *Pseudomonas aeruginosa* ATCC 9721, and *Chromobacterium violaceum* (ATCC 1247). Each bacterial culture was prepared in Luria Berthani (LB) broth/agar and/or Mueller–Hinton broth (MHB).

### 2.7. Minimum Inhibitory Concentration

The minimum inhibitory concentrations (MIC) of a compound against *Streptococcus pyogenes*, *Staphylococcus aureus*, *Escherichia coli*, *Klebsiella pneumoniae*, *Neisseria gonorrhoeae*, and *Pseudomonas aeruginosa* were determined, using the broth dilution method on 96-micro-well plates, as previously described by [17], with slight modifications. A 1 mg/mL solution was prepared for each compound. Briefly, 100 µL of Mueller–Hinton broth (MHB) was transferred in every well and 100 µL of each compound (in triplicate) was transferred into wells in Row A of the micro-titer plate together with the negative (1% dimethyl sulfoxide) and positive controls (ciprofloxacin). Additionally, a blank (sterile MH broth) and standardized bacterium (control) were prepared by transferring 200 µL to the wells, respectively. Two-fold serial dilutions were performed, resulting in decreasing concentrations over the range of 250–1 µg/mL. Thereafter, 100 µL of the standardized bacterium was added into wells of the micro-well plate. After 24-h incubation at 37 °C, 40 µL of *P*-iodonitrotetrazolium (INT, 200 µg/mL) was added and incubated for a further 30 min to 1 h, until an optimal color development. Bacterial growth inhibition (clear wells, no color change) was assessed visually and recorded. The MIC was recorded as the lowest concentration of the extract that inhibited bacterial growth.

### 2.8. Anti-Quorum Sensing Activity of CF6

Anti-quorum sensing activity was tested against the bacterium *Chromobacterium violaceum* ATCC 12472 using the microdilution method, with slight modifications. The positive control used was vanillin/cinnamaldehyde. Before incubation, the absorbance was read at OD_600nm_ (to check the viability and growth of the bacterium) and OD_485nm_ (violacein production). The plates were then incubated at 30 °C for 24 h, shaking at 120 rpm. Following incubation, absorbance was read again at OD_600nm_. Thereafter, the plates were placed in a drying oven at 50 °C for 24 h. After drying, 150 µL of 100% DMSO was used to re-suspend in each well. This was done to confirm that the compound inhibited quorum sensing without influence on bacterial growth activity. It was mixed thoroughly and placed in the shaking incubator at 30 °C, 120 rpm for 1–2 h. Thereafter, absorbance was read at an OD_485nm_ for violacein quantification. The percentage (%) inhibition was determined using Equation (1):(1)$$Percentage\;\left(\%\right)\;inhibition=\frac{\left(OD\;negative\;control-OD\;experimental\right)}{\left(OD\;negative\;control\right)}\times 100\;$$
where OD is the optical density at 485 nm.

### 2.9. AntiBiofilm Assays

#### 2.9.1. Cell Attachment and Biofilm Development

The antibiofilm assay was followed according to [18], with slight modifications. Briefly, the compounds were tested against the six pathogens for both cell attachment and biofilm development inhibition at their respective MIC values (60, 125, and 250 µg/mL). In the cell attachment inhibition assay, 100 μL of standardized bacterial suspension (OD_600nm_ = 0.1), 100 μL of MH broth, and 100 μL of compounds were added to the wells. The positive control (ciprofloxacin 1 µg/mL) and negative control (1% DMSO) were also added into the wells. A volume of 200 μL of sterile MH broth (blank wells) was used and thereafter incubated at 37 °C for 24 h.

Crystal violet (CV) staining procedure: Following incubation, cell attachment was evaluated by the CV staining assay. The wells were washed three times with sterile, distilled water to remove the contents. The remaining biofilm left on the walls of the wells were then oven-dried at 60 °C for 45 min. Following drying, the wells were stained with 100 µL of 1% crystal violet solution and incubated at room temperature for 15 min. The wells were then rinsed three times with sterile, distilled water to remove the excess, unabsorbed stain. To destain the wells, 125 µL of ethanol was added to each well and gently swirled to dissolve the stain from the biofilm. Blank wells were used to zero the microplate reader before taking the OD readings. The absorbance was determined at 585 nm using a SpectraMax Paradigm microplate reader (Molecular devices, Separations, South Africa). The percentage of inhibition was quantified using Equation (1).

#### 2.9.2. Biofilm Development

A standardized bacterial suspension (100 μL) and 100 μL of MH broth were added to the wells and incubated at 37 °C for 8 h. After incubation, 100 μL of extracts and controls were transferred into respective wells and incubated further for 24 h. Biofilm biomass was assessed using the modified crystal violet (CV) assay. The 96-well plates containing formed biofilm were washed with sterile, distilled water to remove planktonic cells and media. The plates were then oven-dried at 60 °C for 45 min. Following drying, 1% CV solution (Merck, Johannesburg, SA) was used to stain the remaining biofilm for 15 min in the dark. The wells were then washed with sterile, distilled water to remove any unabsorbed stain. Semi-quantitative assessment of biofilm formation was performed by adding 125 µL of 95% ethanol to destain the wells. One hundred microliters (100 µL) of the destaining solution were transferred to a new plate and the absorbance (OD 585 nm) was determined using a multi-mode microplate reader (SpectraMax^®^ paradigm). The percentage of inhibition was determined by applying Equation (1).

## 3. Results

### 3.1. Evaluation of the Purity of the Isolated Compound by TLC and UPLC-MS Analysis

After column chromatography, seven different pooled sub-fractions, labelled **CF1–CF7**, were collected and analyzed by 1- and 2-dimension using TLC to check for the purity of the isolated compounds. The seven pooled fractions were dried and weighed as: **CF1** (0.21 g), **CF2** (0.85 g), **CF3** (2.84 g), **CF4** (0.82 g), **CF5** (5.27 g), **CF6** (0.34 g), and **CF7** (0.48 g) (Figure 1). Upon analysis using TLC, **CF6** revealed a compact, single spot with Rf value of 0.18 in CHCl_3_:EtOAc (9:1 *v*/*v*), indicating a pure compound was isolated. To confirm the purity of the seven isolates, 1.5 mg/mL of solution of each isolate was analyzed using a high-resolution UPLC attached with a PDA and MS detector in negative mode. Results of the analysis showed that **CF1**–**CF4** were not as pure, contrary to the TLC results**.** Thus, **C1**–**CF4** were stored for future purification and biological activity evaluation**.** On the other hand, isolates **CF5**, **CF6**, **and CF7** each indicated a single peak with a mass to charge (*m*/*z*) of 321, with **CF6** having the highest purity, of over 95%, considered good from a natural product, especially when detected by photodiode array detector. **CF6** was recrystallized from hexane: benzene (1:1 *v*/*v*) to afford a reddish, amorphous solid.

### 3.2. Elucidation of the Structure of CF6 from Spectrometric and Chromatographic Data

**CF6** was isolated as a reddish, amorphous solid, UV_CHCl3_ λ_max_ 226, 288 nm. IR (KBr) ʋ 3525 (-OH), 1725 (C = O), and 3012 cm^−1^ (Ph-H). The ^1^H NMR (400 MHz, CDCl_3_), Appendix A, and ^13^C NMR (150 MHz, CDCl_3_), Appendix A chemical shifts, multiplicity, and coupling constant are listed in Table 1. HRESIMS (negative ion mode), Appendix A *m*/*z* 321.1343 [M + 5H]^−^ (Calcd for C_20_H_27_O_3_ 315.4300). HRESIMS fragmentation pattern was *m*/*z* 321 [M + 5H]^−^, 261 (C_17_H_26_O_2_), 58(C_3_H_6_O). An Rf value of 0.18 was obtained for **CF6** by TLC using CHCl_3_:EtOAc (9:1 *v*/*v*) as mobile phase.

From the ^13^C NMR experiment (Appendix A) of **CF6**, 20 peaks were auto-detected. Seven saturated carbon signals resonated at 14.00, 14.11, 18.24, 19.30, 21.08, 21.17, and 21.25 ppm. Slightly further down field were carbons thought to be attached to more electronegative groups and placed in chemical environments that were electron rich. These carbons resonated at 29.72, 31.90, and 39.17 ppm. These were then followed by carbons at 45.98 and 64.17 for potential OCH or OCH_2_ groups. There were signals at 95.20, 95.09, 104.75, and 105.78 ppm thought to be -OCO- or -OC- anomeric groups, 128.96 and 129.86 for possible olefinic groups, and the last, a carbon signal at 160.78, for a conceivable carbonyl carbon. To these carbons were attached the corresponding protons that were integrated from the proton spectra (Appendix A). To investigate the presence of quaternary carbons, the HSQC experiment (Appendix A), which reveals protonated carbons, was looked into. This experiment proved the following carbons as quaternary (160.78, 105.78, 104.75, 95.20, and 22.68). The DEPT and the ATP (Appendix A) experiment concurred with the number of quaternary carbons as five, in addition to three methyl carbons (-CH_3_), six methylene carbons (-CH_2_), and three methylene (-CH) carbons. One could envisage that the structure of the isolated compounds could likely be saturated compounds consisting of O-substituted, olefinic, and carbonyl carbon moieties.

The HMBC experiment (Appendix A) was informative regarding how the different moieties would connect to one another. Figure 2b reveals an important HMBC connection between the different moieties of the **CF6**. The diagnostic long-*J* correlations were those that connected the saturated methyl H-20 at 1.69 ppm of the naphtha ring system at the C-10 bridge at 22.68 ppm. A naphtha ring B H-9 proton at 3.04 ppm linked with the vinyl chain moiety C-11 at 128.96 ppm, while its H-11 proton at 7.03 ppm connected with C-9 carbon at 45.98 ppm in a 2*J* coupling pattern. Finally, the 4-hydroxyfuran-2-one unit of the compound linked with the vinyl H-12 proton at 5.41 ppm using its carbonyl carbon at 160.78 ppm to complete the structure of the compound **CF6**. The small *J_H20_-_H1_ or J_H20_-_H9_* of 2.2 Hz conferred the configuration to C-20 methyl group. In addition, a literature search also helped in assigning the configuration to the C-20 methyl group because CF6 has the skeleton of a labdane-type diterpene [19]. **CF6** was, therefore, elucidated as 10-methyl-8-(propan-17-ylidene)naphthalen-9-yl)-11-vinyl-14-hydroxyfuran-16-one (Figure 2c).

### 3.3. Evaluation of the Biological Activity Potentials of 10-Methyl-8-(propan-17-ylidene)naphthalen-9-yl)-11-vinyl-14-hydroxyfuran-16-one

#### 3.3.1. In Vitro Antibacterial Activities

The pure compound **CF6** was evaluated to validate its antibacterial activity against the selected strains, *Escherichia coli*, *Klebsiella pneumoniae*, *Pseudomonas*
*aeruginosa*, *Streptococcus pyogenes*, *Staphylococcus aureus,* and the common urogenital tract infection causing pathogen *Neisseria gonorrhoeae*. The antibacterial (MIC results) activities against these pathogens are shown in Table 2. Based on our findings, **CF6** showed significantly noteworthy activities, with MIC values of 125 µg/mL against Gram-positive pathogens (*S. pyogenes* and *S. aureus*), while the MIC values against the Gram-negative (*E. coli*, *K. pneumonia* and *P. aeruginosa*) were slightly higher (250 µg/mL). The effect on *N. gonorrhoeae* was more potent with MIC = 60 µg/mL. Based on the documented literature, the activity for pure compounds was graded as noteworthy when the MIC was below 10 μg/mL, moderate when between 10 and 100 μg/mL, or low when greater than 100 μg/mL [20]. Gibbons [21] recommends 64 µg/mL as significant MIC for pure or single chemical compounds. For this reason, **CF6** showed significant MIC against *N. gonorrhoeae,* while low MIC values were notable for the other Gram-negative and Gram-positive pathogens. The slightly higher MIC values on the Gram-negative bacteria may be attributed to various factors. Gram-negative bacteria are known to have the outer membrane, efflux pumps, and other components on the cell wall structure, which contribute to resistance to antimicrobials.

#### 3.3.2. In Vitro Antiquorum Sensing Activities of CF6

When **CF6** was evaluated for antiquorum sensing (AQS) potential, results showed no noteworthy inhibitory activity at significantly low concentrations (1–250 µg/mL) (Figure 3). At low concentrations of **CF6**, *Chromobacterium violaceum* was induced to produce increased violacein as opposed to reduction, shown by the negative values of −17.87 ± 0.16 to −6.89 ± 0.32% for **CF6**. **CF6** (250 µg/mL) showed ~51.88% violacein production inhibition. The AQS activity of the **CF6** compound appeared lower compared to that of the AQS-positive controls, cinnamaldehyde and vanillin at 250 µg/mL (Figure 3).

#### 3.3.3. Antibiofilm Activities of **CF6**

Evaluating the antibiofilm activity via the in vitro, static, closed microtiter-based system at cell attachment, the **CF6** compound exhibited improved activity. *S. pyogenes*, *N. gonorrhoea* and *S. aureus* were significantly inhibited from attaching by **CF6** at 80.70%, 81.19%, and 77.62%, respectively (Table 3).

The effect of **CF6** was quite comparable to that of the antibiotic ciprofloxacin (84.33%, 78.24, and 94.52%, respectively). *P. aeruginosa* (16.23%) and *K. pneumonia* (42.92%) were weakly affected by **CF6**.

Contrary to the findings of the **CF6** on cell attachment, biofilm development was weakly disrupted for all the pathogens. *E. coli* biofilm development was slightly induced (−4.65%) in the presence of **CF6**. The inhibitory effect of **CF6** on the biofilm development ranged between 29.08–35.55%. Although **CF6** was not effective, the ciprofloxacin was, likewise, weakly effective (40.72–48.31%) in disrupting the formation of the biofilm. The above findings were not unanticipated due to the fact that formed biofilms are more complex and difficult to disrupt; hence, it is rather easier to prevent the biofilm formation.

## 4. Discussion

The structural analogues of **CF6** were previously reported from other medicinal plants by [22] and their biological activities were reviewed in depth by [23]. The non-polar extracts of *H. caespititium,* from where **CF6** was isolated, exhibited antioxidant and antimicrobial activities [16]. However, the hexane extract demonstrated the best anti-gonorrhea activity against *Neisseria gonorrhea* reference strains as well as being least toxic against rat liver cells. We strongly proposed that **CF6** would exhibit a comparative anti-gonorrheal and other antimicrobial activities like the non-polar extracts of *H. caespititium.*

Although the observed activities were considerably noteworthy, their effect was not as comparable to that of amoxicillin and ciprofloxacin (MIC = 1 µg/mL, respectively). The anti-bactericidal effect of ciprofloxacin was attributed to targeting the DNA replication inhibition. Ciprofloxacin has a broad-spectrum activity, affecting mostly the Enterobacteriaceae such as *Escherichia coli*, *Salmonella* spp., *Shigella* spp., and *Neisseria* [24]. Since our **CF6** compound demonstrated proficiency on both the GNB and GPB pathogens, we can reason that it has a broad-spectrum activity, making it a desirable potential for drug discovery, though the mechanism of action is not known. The added sensitivity of GPB to **CF6** may be attributed to the easy access to the thick bacterial cell wall component.

Based on reported literature, *H. caespititium* is known for its health beneficial properties due to their phytoconstituents. Its plant extracts are well documented for antibacterial, antigonorrhoeic, antimycobacterial, antifungal, and other biological activities (for a detailed review, see [25]). The potent inhibitory properties against many pathogens are majorly attributed to the active compound 2-(4-methylpentanoyl)-4(3-methylbuten-2-yl)-phloroglucinol (caespitin) [11]. Van der Schyf [26] evaluated the *H. caespititium* extract against *S. pyogenes*, *Proteus mirabilis*, *Pseudomonas aeruginosa*, *Escherichia coli*, and *S. aureus*. The findings showed a potent antibacterial activity with a MIC value of 80 µg/mL against the GPB (*S. pyogenes* and *S. aureus*). The reported activities [21] were lesser compared to our findings, where MIC values against *S. pyogenes* and *S. aureus* were found to be 125 µg/mL, yet the inhibitory concentration was still comparatively potent. Kutluk and colleagues [27] reported antibacterial activities of *Helichrysium* species (*H. araxinum* Takht. ex Kirp., *H. armenium* DC, *H. arenarium*, *H. pallasii*, *H. stoechas*, *H. sanguineum*, and *H. graveolens*) against GNB (*E. coli*, *P. aeruginosa*, *P*. *mirabilis*, *K. pneumoniae*, *A. baumanni*) at 32–64 µg/mL concentrations, while the GPB were affected at 8-64 µg/mL. Mamabolo [16] reported the non-polar dichloromethane extract of *H. caespitium* has significant antimicrobial properties, particularly antigonorrheal and antioxidant activities. Mamabolo et al. [16] reported the whole plant of the *H. caespititium n-*hexane extract as active against four *N. gonorrhoea* strains with MIC values ranging from 37–330 µg/mL. Our compound, **CF6**, showed significant MIC results of 60 µg/mL against *N. gonorrhoea*, and, hence, **CF6** antigonorrhoeic potential. Whereas Mamabolo and co-workers evaluated only the extracts, the present study evaluated the single chemical entity (**CF6**) from the plant *H. caespititium*.

The induction of violacein at low concentrations may be attributed to the bacterium utilising the compound as a supplement of growth or violacein production provocation. At this point and based on documented literature, there is no direct correlation between the violacein production and growth of the bacterium. The violacein, an indicator of quorum sensing, is a natural pigment produced by *Chromobacterium violaceium*. Violacein [3-(1, 2-dihydro-5-(5-hydroxy-1*H*-indol-3-yl)-2-oxo-3*H*-pyrrol-3-ilydene)-1,3-dihydro-2*H*-indol-2-one] appears as the purple colonies on the bacterial growth media (agar) and is known for bactericidal, trypanocidal, tumoricidal, mycobactericidal, and antioxidant activities [28]. According to Ahmad et al. [28], some plant materials have potential to contribute to increased violacein or pigment production yield due to the presence of sugars and L-tryptophan in the plant material.

Based on the literature search, there were no available data or information of the anti-quorum sensing activity of **CF6** or for the source plant, *H. caespititum*. This study reports, for the first time, the antiquorum sensing potential of **CF6**. The AQS potential may be attributed to the binding potential to the receptor protein of *Chromobacterium violaceium*, where the compounds are potentially competing for the active binding site.

Further assessment of biological activities of **CF6** and antibiofilm (cell attachment and biofilm development) was considered. This is because the cell attachment is vital, whereby a single cell bacterium adheres to another, and/or the extracellular matrix, and/or to the surface. This is also considered as microbial colonies aggregated together and attached to surfaces or without attaching to surfaces [29]. The initial step in vitro (static) was characterized by the cells attaching together, forming the subsequent bonds, and promoting quick proliferation, thereby increasing the total adhesion strength [30]. Time significantly contributes to the strengthening of the attachment, and, thereafter, the formation of a mature biofilm, resistant to treatment or disruption.

In a similar manner to the antibacterial findings, **CF6** was able to significantly prohibit the cell attachment of *N. gonorrhoea* followed by *S. pyogenes* and *S. aureus*. Based on the literature search, there is a paucity of information on the prevention of biofilm formation or attachment activity of *H. caespititum*. This will be the first time reporting compound **CF6** with potential to prohibit biofilm formation or cell-to-cell attachment. The potential mechanism of action for inhibiting cell attachment may be attributed to the compound disturbing the cohesiveness of extracellular polysaccharide (EPS) of bacterial origin by reducing EPS cohesiveness. Alternately, the compound affects the EPS and/or destroys the cells prior to cell attachment.

## 5. Conclusions

It is pertinent to mention that 10-methyl-8-(propan-17-ylidene)naphthalen-9-yl)-11-vinyl-14-hydroxyfuran-16-one and its biological activities, herein reported, validated the anti-gonorrheal mitigating properties of the non-polar extracts of South African *Helichrysum caespititium*. As a result, the South African Indigenous Knowledge Systems (IKS) practitioners may familiarize themselves with the method of extraction that this study described. Furthermore, re-purification and biological evaluation of **C1**–**C4** is underway in our laboratory. Their antibacterial and, in particular, anti-gonorrheal activity may further validate *H. caespititium* as an antigonorrheal plant drug.

## Figures and Tables

**Figure 1 biology-10-01224-f001:**
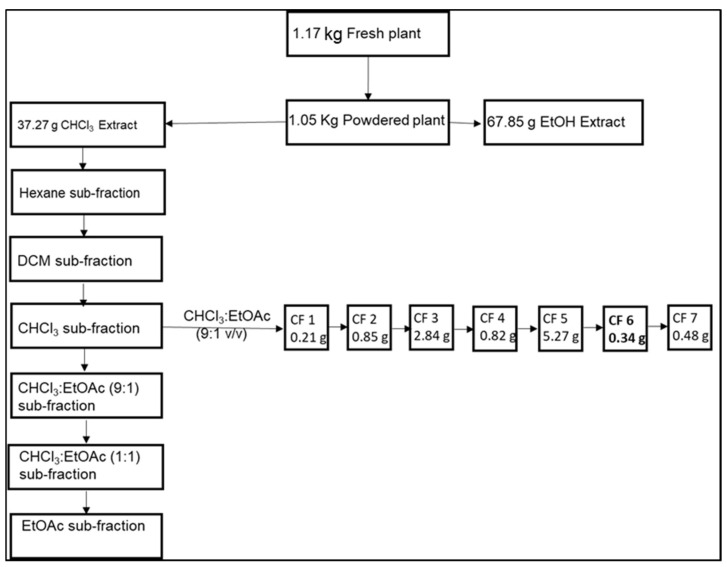
The summary of the isolated compounds is displayed.

**Figure 2 biology-10-01224-f002:**
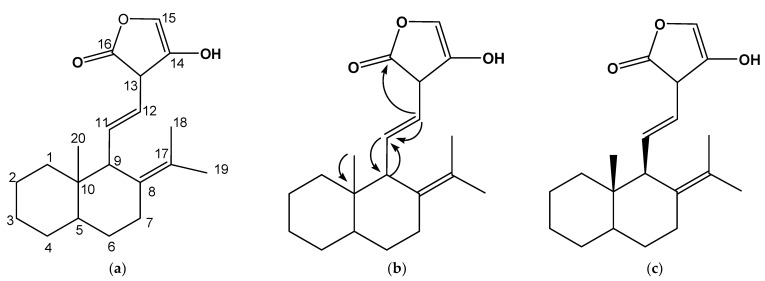
The numbered skeleton (**a**), the assigned HMBC (**b**), and the fully elucidated structure of compound 10-methyl-8-(propan-17-ylidene)naphthalen-9-yl)-11-vinyl-14-hydroxyfuran-16-one (**CF6**) (**c**).

**Figure 3 biology-10-01224-f003:**
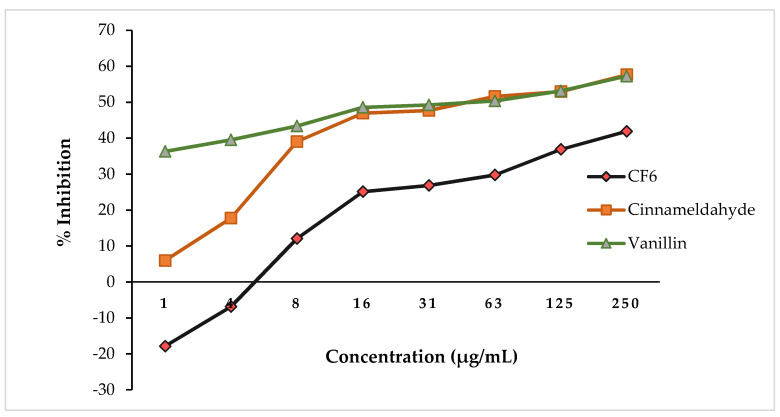
**CF6** vs. cinnamaldehyde and vanillin showing anti-quorum sensing potential in a concentration-dependent manner (1–250 µg/mL).

**Table 1 biology-10-01224-t001:** ^1^H NMR (400 MHz,) and ^13^CNMR (150 MHz) data for CF6 in CDCl_3_.

Position	CF6			
	Assignment	^δ^ _H_	Multiplicity (*J*/Hz)	^δ^ _C_
1	CH_2_	2.11	t(2H, *J* = 2.2 Hz)	21.08
2	CH_2_	3.43	d(2H, *J =* 6.8 Hz)	21.17
3	CH_2_	1.79	m(2H)	19.37
4	CH_2_	1.72	m(2H)	21.25
5	CH	3.92	m(H)	39.17
6	CH_2_	1.24	m(2H)	29.72
7	CH_2_	2.17	t(2H, *J* = 2.1 Hz)	31.90
8	Cq	-	-	105.78
9	CH	3.04	t(H, *J* = 7.2 Hz)	45.98
10	Cq	-	-	22.68
11	C = C-H	7.03	dd(H, *J* = 10.2, 16.0)	128.96
12	C = C-H	5.41	dd(H, *J =* 8.0 Hz)	129.86
13	CH	4.75	m(1H)	64.17
14	OCq	-	-	95.09
15	OCH	5.89	s(H)	95.20
16	O = Cq	-	-	160.78
17	Cq	-	-	104.75
18	CH_3_	0.96	s(3H)	14.00
19	CH_3_	0.96	s(3H)	14.11
20	CH_3_	1.69	s(3H))	18.24

Cq is the quaternary carbon.

**Table 2 biology-10-01224-t002:** Minimum inhibitory concentration (MIC in µg/mL) for the isolated compounds against six pathogens.

Compounds	*S. pyogenes*	*S. aureus*	*E. coli*	*K. pneumoniae*	*P. aeruginosa*	*N. gonorrhoeae*
**CF6**	125	125	250	250	250	60
**Controls** (Ciprofloxacin = 1 µg/mL for all test pathogens)						

The same MIC values were observed in triplicate results on two separate occasions.

**Table 3 biology-10-01224-t003:** Inhibition of cell attachment and biofilm development (at respective MIC values) on selected pathogens following exposure to **CF6**.

**Cell Attachment (%)**						
**Compounds**	** *S. pyogenes* **	** *S. aureus* **	** *E. coli* **	** *K. pneumoniae* **	** *P. aeruginosa* **	** *N. gonorrhoeae* **
**CF6**	80.70 ± 0.10	77.62 ± 0.07	65.89 ± 0.23	40.76 ± 0.17	42.03 ± 0.39	81.19 ± 0.11
Ciprofloxacin	84.33 ± 0.04	78.24 ± 0.02	76.80 ± 0.08	78.58 ± 0.19	79.79 ± 0.04	94.52 ± 0.01
**Biofilm Development (%)**						
**Compounds**	** *S. pyogenes* **	** *S. aureus* **	** *E. coli* **	** *K. pneumoniae* **	** *P. aeruginosa* **	** *N. gonorrhoeae* **
**CF6**	29.08 ± 0.01	35.55 ± 0.14	−4.65 ± 0.19	25.64 ± 0.19	25.28 ± 0.35	32.57 ± 0.35
Ciprofloxacin	42.89 ± 0.04	42.36 ± 0.08	42.91 ± 0.03	43.62 ± 0.14	48.31 ± 0.07	40.72 ± 0.11

## Data Availability

Not applicable.

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
