# Peer review of "An Andrographolide from Helichrysum caespitium (DC.) Sond. Ex Harv., (Asteraceae) and Its Antimicrobial, Antiquorum Sensing, and Antibiofilm Potentials"

_biology, 2021, doi:10.3390/biology10121224_

Round 1

Reviewer 1 Report

  • Biology-1409985-peer-review-v1
  •  
  • The paper has significant weaknesses in its presentation.
  •  
  • Assignment of beta settings are weak, relevant bibliography should be incorporated.

On the other hand, beta configuration is assigned to the methyl of position 20? By authors, however a configuration alpha is reported in the IUPAC name.

The compound should be numbered, for a better reading.

Please see Chem. Pharm. Bull. 56(2) 210—212 (2008); https://doi.org/10.1248/cpb.56.210

The small JH20-H1 or JH20-H9 of 2.2 Hz confers the β-configuration to C-20 methyl group. In addition, literature search also help in as- signing the -configuration to the C-20 methyl group because CF6 has the skeleton of a 271“normal” labdane [11].

Please revise IUPAC Name:

“CF6 was therefore elucidated as 3-((E)-2-((8αS)-decahydro-8α-me- 272thyl-2-(propan-2-ylidene)naphthalen-1-yl)vinyl)-4-hydroxyfuran-2(3H)-one, Figure 2c.”

  • About 37.27 g of H. caespititium chloroform extract dissolved in 10.0 mL of chloroform was adsorbed to 35.01 g of dry silica and allowed to dry.

The process of dissolving 37 grams of an extract in 10 milliliters is not possible. This should be checked by the authors, maybe there is a typing error. Nor is it possible to adsorb those grams in the same amount of silica, it would not meet the basic criteria of the relationship between sample and silica necessary for a correct separation. Please check your calculations.

A column with fifty grams of silica would not allow 70 grams of a sample to be separated (37.27 g of H. caespititium chloroform extract dissolved in 10.0 mL of chloroform was adsorbed to 35.01). It's not possible.

  • Section 2.4. Column Chromatography

Please include in Figure 1 the yield for each fraction and subfraction, as reported the data is very preliminary or incomplete.

  • Section 3.1. Evaluation of the Purity of the Isolated Compound by TLC and UPLC-MS Analysis

Lines 229-223, the authors write: “ On the other  hand, isolates CF5, and CF7 indicated the same mass to charge (m/z) of 321 [M-H] with  CF6 having the highest purity of over 95%, considered good from a natural product especially when detected by photodiode array detector. CF6 was recrystallized from hexane: benzene (1:1 v/v) to afford a reddish amorphous solid.”

However in section 3.2. Elucidation of the Structure of CF6 From Spectrometric and Chromatographic Data, the authors write

HRESIMS (negative ion mode) m/z 321.1343 [M+5H]- (Calcd for C20H27O3 315.4300). 240 HRESIMS fragmentation pattern are m/z 321 [M+5H]-, 261 (C17H26O2), 58(C3H6O)

321 [M-H] or 321.1343 [M+5H]-

  • The assignment of the “E” system of the hydrogens of carbons 11 and 12 should be revised. It is a clean system, and it is expected signs and constant links, which show your neighborhood.

Please see Chem. Pharm. Bull. 56(2) 210—212 (2008)

Cytotoxicity of Labdane-Type Diterpenoids from Hedychium forrestii

Qing ZHAO, Chen QING, Xiao Jiang HAO, Jun HAN, Guo Ying ZUO, Cheng ZOU, and Gui Li XU. https://doi.org/10.1248/cpb.56.210.

  • Spectra of supplementary material are not available, please check.

The manuscript in its current state should not be accepted.

Author Response

Line/page

REVIEWER 1 SUGGESTIONS

AUTHORS RESPONSE

All document

The paper has significant weaknesses in its presentation.

The authors acknowledge the reviewers concerns regarding the paper and thus revise as per suggestions

All document

Assignment of beta settings are weak, relevant bibliography should be incorporated

b-configuration has been revised and assigned to C-10 in CF6 and bibliography included as suggested.

All document

On the other hand, beta configuration is assigned to the methyl of position 20? By authors, however a configuration alpha is reported in the IUPAC name.

The IUPAC name of CF6 has been revised

L280-282

The compound should be numbered, for a better reading.

There is only one compound, (CF6) the its numbered in Figure 2a.

Please see Chem. Pharm. Bull. 56(2) 210—212 (2008); https://doi.org/10.1248/cpb.56.210

Reference was consulted and helped in revising the IUPAC name of CF6

L274/275

The small JH20-H1 or JH20-H9 of 2.2 Hz confers the β-configuration to C-20 methyl group. In addition, literature search also help in as- signing the b-configuration to the C-20 methyl group because CF6 has the skeleton of a 271“normal” labdane [11].

Revised to……” CF6 has the skeleton of a  labdane-type diterpen

L275/276

Please revise IUPAC Name:

“CF6 was therefore elucidated as 3-((E)-2-((8αS)-decahydro-8α-me- 272thyl-2-(propan-2-ylidene)naphthalen-1-yl)vinyl)-4-hydroxyfuran-2(3H)-one, Figure 2c.”

The IUPAC name of CF6 has been revised to 10b-methyl-8-(propan-17-ylidene)naphthalen-9-yl)vinyl-14-hydroxyfuran-16-one

L34

·        About 37.27 g of H. caespititium chloroform extract dissolved in 10.0 mL of chloroform was adsorbed to 35.01 g of dry silica and allowed to dry.

Revised to 37.27 g was concentrated in 50 mL of chloroform.

This was for ADSORPTION of the extract to dry silica. This happens prior to loading to the column.

The process of dissolving 37 grams of an extract in 10 milliliters is not possible. This should be checked by the authors, maybe there is a typing error. Nor is it possible to adsorb those grams in the same amount of silica, it would not meet the basic criteria of the relationship between sample and silica necessary for a correct separation. Please check your calculations

Its not dissolving but ADSORBING prior to loading the sample to a packed glass column.

Small amount of dry silica is placed in a mortar and the highly concentrated extract solution(hence 50 ml vs 37.27 g extract) is added in drops and the mixture is sort of blended (smeared) with pestle and allowed to dry before loading a packed column and the mobile phase added.

L116

L119 - 120

A column with fifty grams of silica would not allow 70 grams of a sample to be separated (37.27 g of H. caespititium chloroform extract dissolved in 10.0 mL of chloroform was adsorbed to 35.01). It's not possible.

Column dimension used is stated as:

A 30 mm od x 2.0 mm wall x 600 mm long.

The column was wet packed with a silica gel slurry prepare by adding 50 g of dry silica in 50 mL of acetone. The silica slurry was filled up to 65% the length of the column was used in this study.

50 g versus 70 g sample is not mentioned in the manuscript

37.27 g extracts in 10 mL

Figure 1 and  L68/71

·        Section 2.4. Column Chromatography

Please include in Figure 1 the yield for each fraction and subfraction, as reported the data is very preliminary or incomplete.

Figure 1 has the yield of the chloroform sub-fraction reported in (g)

Those of main faction were not recorded as we knew our target come is in the chloroform sub-fraction based on previous study as stated in Line 68 - 71

L233 - 234

·        Section 3.1. Evaluation of the Purity of the Isolated Compound by TLC and UPLC-MS Analysis

 Lines 229-223, the authors write: “ On the other  hand, isolates CF5, CF6 and CF7 indicated the same mass to charge (m/z) of 321 [M-H] with  CF6 having the highest purity of over 95%, considered good from a natural product especially when detected by photodiode array detector. CF6 was recrystallized from hexane: benzene (1:1 v/v) to afford a reddish amorphous solid.”

Revised to:

isolates CF5, CF6 and CF7 each indicated a single peak with a mass to charge (m/z) of 321with  CF6 having the highest purity of over 95%,

L244 - 245

However in section 3.2. Elucidation of the Structure of CF6 From Spectrometric and Chromatographic Data, the authors write

 HRESIMS (negative ion mode) m/z 321.1343 [M+5H]- (Calcd for C20H27O3 315.4300). 240 HRESIMS fragmentation pattern are m/z 321 [M+5H]-, 261 (C17H26O2), 58(C3H6O), 321 [M-H] or 321.1343 [M+5H]-

Seemingly no suggestion from reviewer

321.1343 is the exact mass of CF6 + 5H adduct

315.4300 is calculated from the elucidated molecular structure of C20H27O3.

Daughter ions (fragmentation patterns as:

HRESIMS fragmentation pattern are m/z 321 [M+5H]-, 261 (C17H26O2), 58(C3H6O)

·        The assignment of the “E” system of the hydrogens of carbons 11 and 12 should be revised. It is a clean system, and it is expected signs and constant links, which show your neighborhood.

Please see Chem. Pharm. Bull. 56(2) 210—212 (2008)

Cytotoxicity of Labdane-Type Diterpenoids from Hedychium forrestii

Qing ZHAO, Chen QING, Xiao Jiang HAO, Jun HAN, Guo Ying ZUO, Cheng ZOU, and Gui Li XU. https://doi.org/10.1248/cpb.56.210.

 Spectra of supplementary material are not available, please check

H-11 and H-12 has been revised as indicated on the revised IUPAC nomenclature of CF6. Chem. Pharm. Bull. 56(2) 210—212 (2008) assisted in this regards.

Consulted

Consulted

Spectra were submitted to the Editor

Reviewer 2 Report

I read with interest the manuscript and I believe that the overall quality of the work is good, the novelty is quite high and the used protocol adequate to support results. 

I believe that the manuscript could be accepted after a revision made according to reviewers' suggestions.

Here, my comments. 

1) Check botanicals names and revise them including the botanical descriptors at the first citation: for example:

Helichrysum caespititium (DC.) Sond. Ex Harv. 

 2) Edit and revise text in order to avoid underlined text and other minor errors. 

3) In the introduction, or better at the beginning of 3.3, please describe in a clearer way why authors chose to focus on CF6 and not on other compounds of non-polar extract.

4) 2.7: ATCC numbers could be avoided here, as they are reported above.

5) 2.9.1: I believe that a better description of the method is required for the cell attachment assay.

6) Table 2: Did authors obtain the same results in all replicates? Please add a sentence to give this detail.

7) 3.3.2 Revise and write this part in a clearer way.

Author Response

Line/page

REVIEWER 2 SUGGESTIONS

AUTHORS RESPONSE

All document

I believe that the manuscript could be accepted after a revision made according to reviewers' suggestions.

The authors appreciate the reviewer and their time and effort to review this paper. As such, we are happy to revise the paper as per reviewer’ suggestions.

Title

1) Check botanicals names and revise them including the botanical descriptors at the first citation: for example:

Helichrysum caespititium (DC.) Sond. Ex Harv. 

Revised as suggested

2) Edit and revise text in order to avoid underlined text and other minor errors. 

Done

L231 - 232

3) In the introduction, or better at the beginning of 3.3, please describe in a clearer way why authors chose to focus on CF6 and not on other compounds of non-polar extract.

Already mentioned in Lines 231- 232

Results of the analysis showed that CF1 - CF4 were not as pure contrary to the TLC results. Thus, C1 – CF4 were stored for future purification and biological activity evaluation. CF6 was the purest to be investigated for its biological potentials

4) 2.7: ATCC numbers could be avoided here, as they are reported above.

The ATCC number have been removed as per reviewer’s suggestion.

5) 2.9.1: I believe that a better description of the method is required for the cell attachment assay.

Additional information of the method has been incorporated in the respective section as suggested by the reviewer.

6) Table 2: Did authors obtain the same results in all replicates? Please add a sentence to give this detail.

The MIC values were read in triplicates on two separate occasions and same concentrations were observed. This has been stated as such under Table 2.

7) 3.3.2 Revise and write this part in a clearer way.

Section 3.3.2 has been revised for better understanding.

Reviewer 3 Report

The paper of Kokoette Edward Bassey, Patience Mamabolo and Sekelwa What entitled: "An Andrographolide from Helichrysum caespitium and its An-1 timicrobial, Antiquorum Sensing and Antibiofilm Potentials" is particularly interesting for the contribution it provides to research on the various potential biological activities concerning different genera and species belonging to the Asteraceae family, and not only for its recognized antimicrobial activity. For this reason some suggestions would be useful:

  • Authors should insert in the title the name of the family next to the name of the species studied as follows: “An Andrographolide from Helichrysum caespitium (Asteraceae) and its An-1 timicrobial, Antiquorum Sensing and Antibiofilm Potentials”;
  • In the Abstract and in the Introduction, the name of the species should be followed by the name of the authors as follows: Helichrysum caespititium (DC.) Harv.;
  • Authors should add the abbreviation point to the authors in the text as follows: DC. and Harv.;
  • To give greater emphasis to the potential pharmaceutical interest of the Asteraceae family, the Authors could consider and then add, for example, the following references:
  • Chiavari-Frederico MO.; Barbosa LN.; Carvalho Dos Santos I.; Ratti da Silva G.; Fernandes de Castro A.; de Campos Bortolucci W.; Barboza LN.; Campos CFAA.; Gonçalves JE.; Menetrier JV.; Jacomassi E.; Gazim ZC.; Wietzikoski S.; Dos Reis Lívero FA.; Wietzikoski Lovato EC. Antimicrobial activity of Asteraceae species against bacterial pathogens isolated from postmenopausal women.  PLoS One 2020, 15(1), e0227023. doi: 10.1371/journal.pone.0227023.
  • Freitas PR.; de Araújo ACJ.; Dos Santos Barbosa CR.; Muniz DF.; Rocha JE.; de Araújo Neto JB.; da Silva MMC.; Silva Pereira RL.; da Silva LE.; do Amaral W.; Deschamps C.; Relison Tintino S.; Ribeiro-Filho J.; Coutinho HDM. Characterization and antibacterial activity of the essential oil obtained from the leaves of  Baccharis coridifolia DC against multiresistant strains. Microb Pathog 2020, 145, 104233. doi: 10.1016/j.micpath.2020.104223.
  • García-Risco MR.; Mouhid L.; Salas-Pérez L.; López-Padilla A.; Santoyo S.; Jaime L.; Ramírez de Molina A.; Reglero G.; Fornari T. Biological Activities of Asteraceae (Achillea millefolium and Calendula officinalis) and Lamiaceae (Melissa officinalis and Origanum majorana) Plant Extracts. Plant Foods Hum Nutr 2017, 72(1), 96-102.
  • Naeim H.; El-Hawiet A.; Abdel Rahman RA.; Hussein A; El Demellawy MA.; Embaby AM.BMC. Antibacterial activity of Centaurea pumilio L. root and aerial part extracts against some multidrug resistant bacteria. Complement Med Ther. 2020, 20(1), 79. doi: 10.1186/s12906-020-2876.
  • Oppedisano F.; Muscoli C.; Musolino V.; Carresi C.; Macrì R.; Giancotta C.; Bosco F.; Maiuolo J.; Scarano F.; Paone S.; Nucera S,; Zito MC.; Scicchitano M.; Ruga S,; Ragusa M.; Palma E.; Tavernese A.; Mollace R.; Bombardelli E.; Mollace V. 2020. The Protective Effect of Cynara cardunculus Extract in Diet-Induced NAFLD: Involvement of OCTN1 and OCTN2 Transporter Subfamily. Nutrients 2020, 12(5,  1435.
  • Panda SK.; Luyten W. Antiparasitic activity in Asteraceae with special attention to ethnobotanical use by the tribes of Odisha.  India Parasite 2018, 25, 10.
  • Rolnik A,; Olas B. The Plants of the Asteraceae Family as Agents in the Protection of Human Health Int J Mol Sci. 2021, 22(6), 3009.
  • Sharifi-Rad M,; Mnayer D.; Morais-Braga MFB.; Carneiro JNP.; Bezerra CF.; Coutinho HDM.; Salehi B.; Martorell M.; Del Mar Contreras M.; Soltani-Nejad A.; Uribe YAH.; Yousaf Z.; Iriti M.; Sharifi-Rad J. Echinacea plants as antioxidant and antibacterial agents: From traditional medicine to biotechnological applications.  Phytother Res 2018, 32(9), 1653-1663.

Author Response

Line/page

REVIEWER 3 SUGGESTIONS

AUTHORS RESPONSE

Title

·        Authors should insert in the title the name of the family next to the name of the species studied as follows: “An Andrographolide from Helichrysum caespitium (Asteraceae) and its An-1 timicrobial, Antiquorum Sensing and Antibiofilm Potentials”;

Authors have incorporated the family names as suggested by the reviewer.

L24, L51

In the Abstract and in the Introduction, the name of the species should be followed by the name of the authors as follows: Helichrysum caespititium (DC.) Harv.

This has been incorporated as suggested by the reviewer. Page 1, line 24. Page 2, line 51.

·        Authors should add the abbreviation point to the authors in the text as follows: DC. and Harv.;

This has been incorporated as suggested by the reviewer. Page 1, line 24. Page 2, line 51.

To give greater emphasis to the potential pharmaceutical interest of the Asteraceae family, the Authors could consider and then add, for example, the following references:

·         

The suggested references have been added.

Round 2

Reviewer 1 Report

The authors have improved their work, however the answer on the structural elucidation of the compound is still weak. The system's coupling constants are unclear and unspecified. This shortcoming does not allow the presented structure to be proposed safely.

The coupling constants (j) are not as expected to trans olefins, according to the bibliography, and neither is the multiplicity of the signals.

The spectra submitted are not very good, and will not show multiplicity of signals or coupling constants.

Please see spectroscopic date to compounds 11 and 12 in Chem Pharm Bull, 42(6), 1216-1225, 1994 Cell differentiation-inducing diterpenes from Andrographis paniculata NeesT MATSUDA, M KUROYANAGI… - Chemical and …, 1994 - jstage.jst.go.jp. https://doi.org/10.1248/cpb.42.1216.

Please , also revise H on C9

The constants between H (C9) and H (C11) should be shown, since they are neighbors and are coupled, the same for H (C11) and (HC12) of the trans system. Please revise https://doi.org/10.1248/cpb.56.210.

Round 3

Reviewer 1 Report

The authors have made the suggested changes, after which the paper should be accepted for publication.

Author Response

The authors would like to thank the reviewer for the time that was invested in reviewing this work.